



# Finding Structure-Property Relationships for Molecular Property Predictions with Globally Explainable AI

Jonas Teufel[1], Pascal Friederich[1,2]

[1] Institute of Theoretical Informatics (ITI), Karlsruhe Institute of Technology (KIT), Kaiserstraße 12, Karlsruhe Germany
[2] Institute of Nanotechnology (INT), Karlsruhe Institute of Technology (KIT), Kaiserstraße 12, Karlsruhe Germany

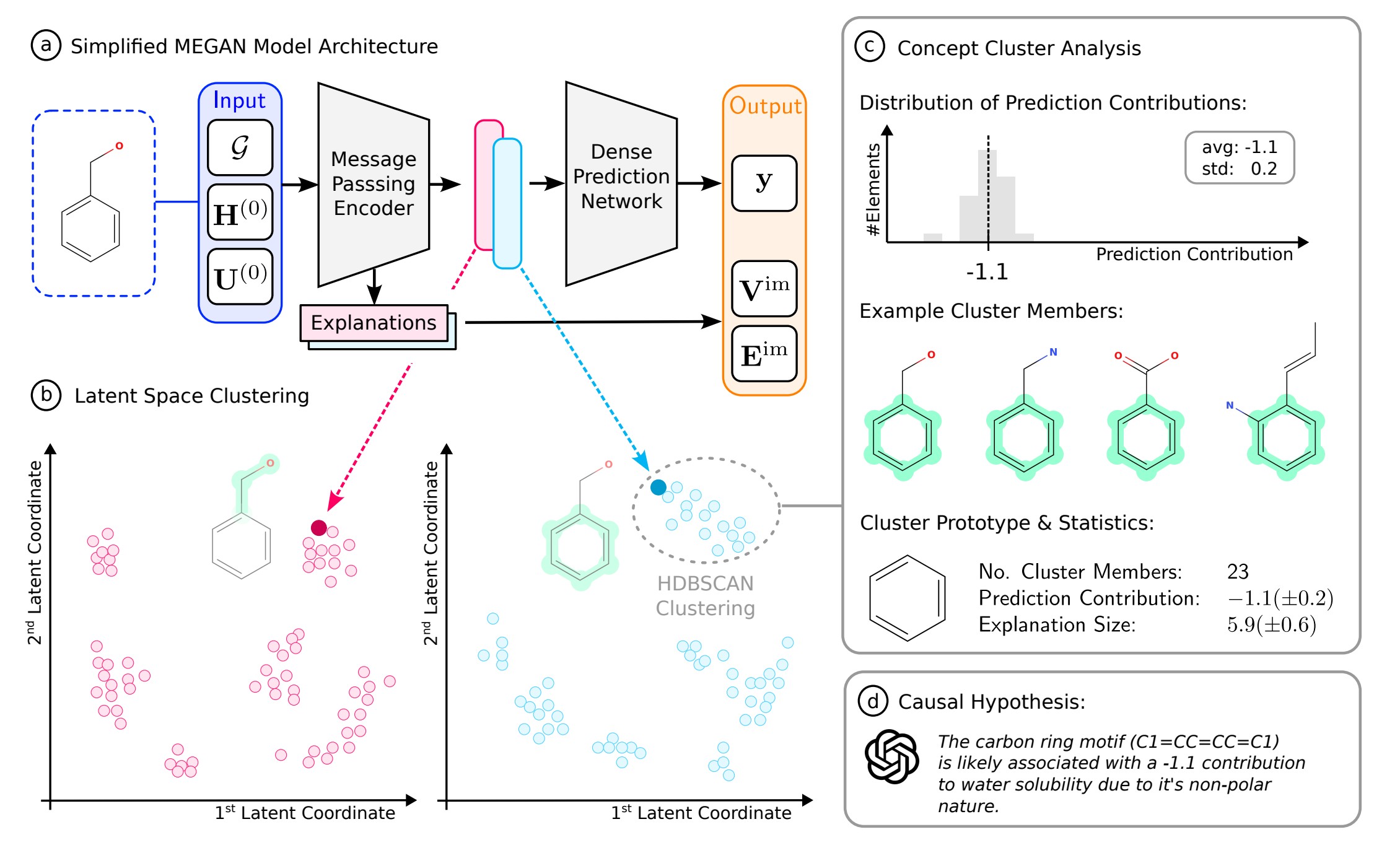

## Motivation

- Powerful AI models have become a useful tool for various predictive and generative tasks.

- For example, *Graph Neural Networks (GNNs)* can be effectively used for various molecular property prediction tasks in chemistry and material science.

- By understanding the internal behavior of high-performing models we can learn about the *structure-property relationships* of the underlying tasks.

- Novel insights into the underlying rule behind certain molecular properties can ultimately help us with *drug discovery* and *material design*.

- We can gain understanding of a model's inner workings through various *Explainable AI (xAI)* methods.

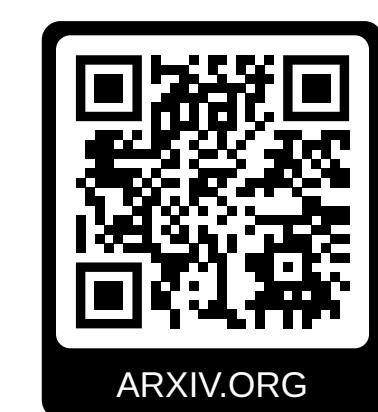

## Extract Scientific Insights from High-Performing AI Models

### a. Explainable MEGAN Model

- Train *Multi-Explanation Graph Attention Network (MEGAN)* for molecular property prediction.

- Model creates *Local Explanation Masks* directly alongside main target prediction.

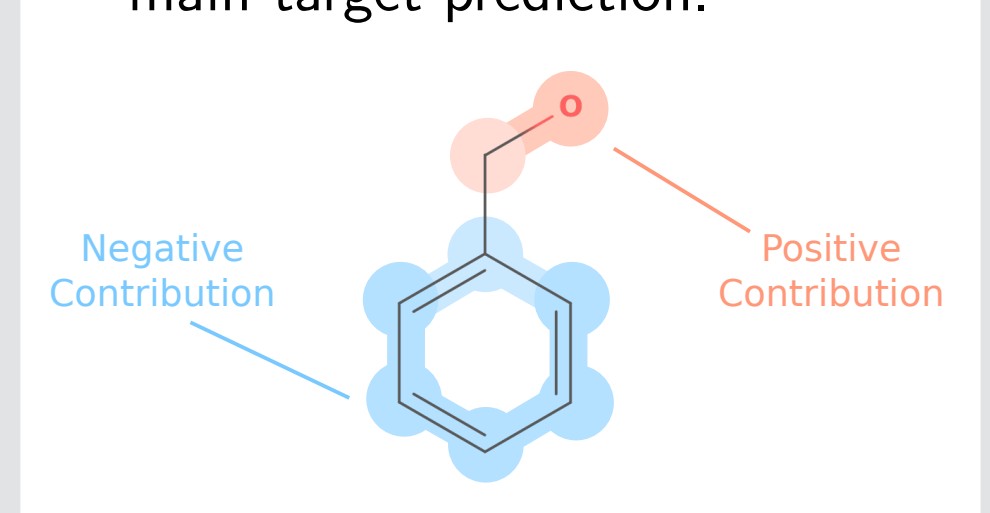

### b. Clustering Latent Explanations

- Contrastive learning objective: latent space similarity ⇒ structural similarity of subgraph motifs.

- Latent space clusters ⇒ elements with similar local explanation.

- HDBSCAN clustering to find dense clusters of elements in latent space.

### c. Analyzing Concept Clusters

- Each explanation embedding is associated with an average contribution towards the final prediction outcome.

⇒ Associating *Structure* (cluster motif) with *Property* (average contribution).

- *Genetic Algorithm* finds a small yet representative *Prototype Graph* for each cluster.

### d. LLM-based Causal Hypothesis

- Convert representative prototype graph for each cluster into string *SMILES* representation.

- Prompt language model (e.g. GPT-4) with prototype SMILES and average contribution.

⇒ Language model creates a hypothesis about a possible causal reason behind the observed structure-property relationship.

## Global Concept Explanations for Graph Neural Networks...

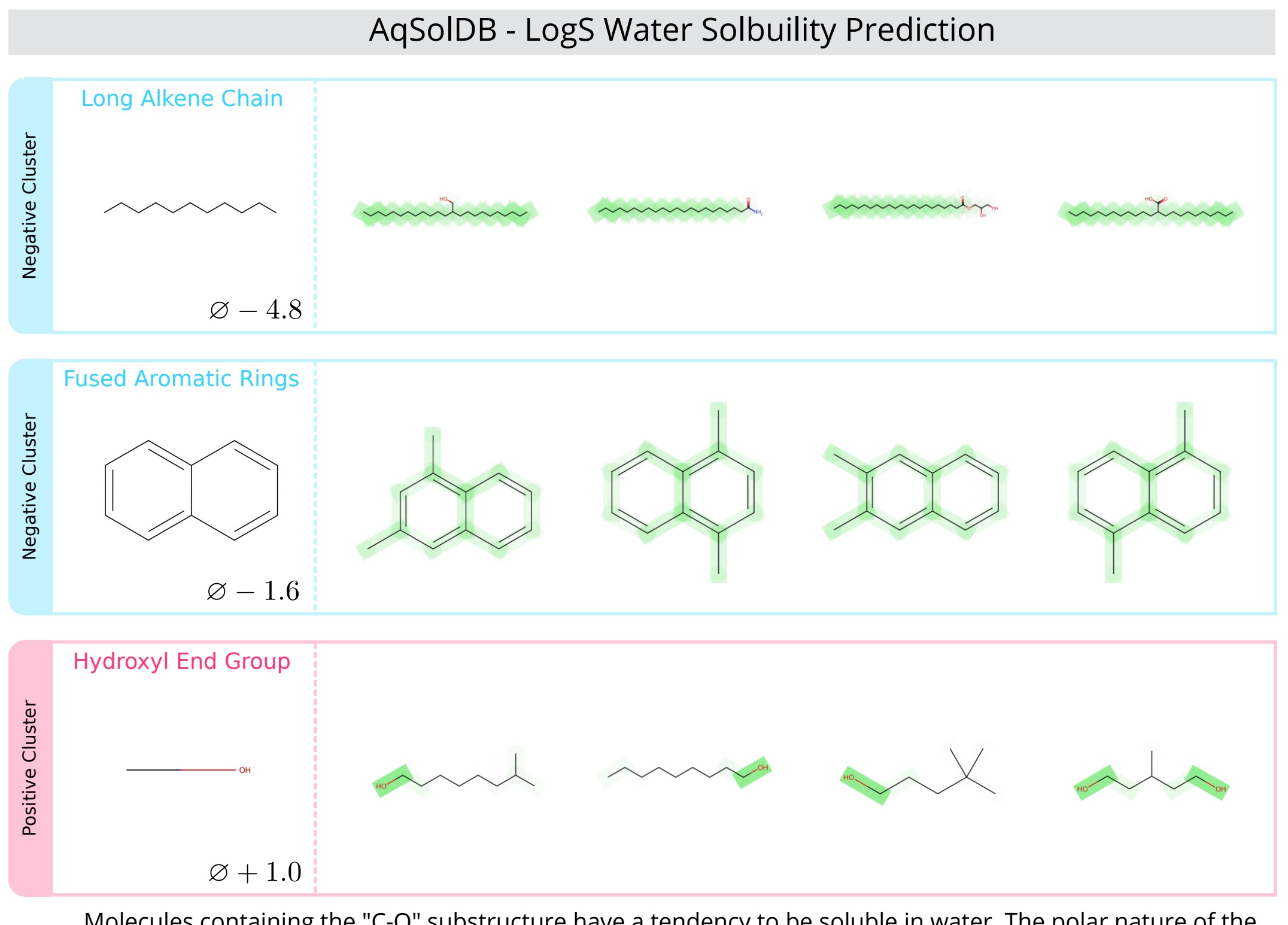

AqSolDB - LogS Water Solubility Prediction

Molecules containing the "C-O" substructure have a tendency to be soluble in water. The polar nature of the carbon-oxygen bond and the ability to form hydrogen bonds with water molecules are hypothesized to be the driving forces behind the high influence on water solubility.

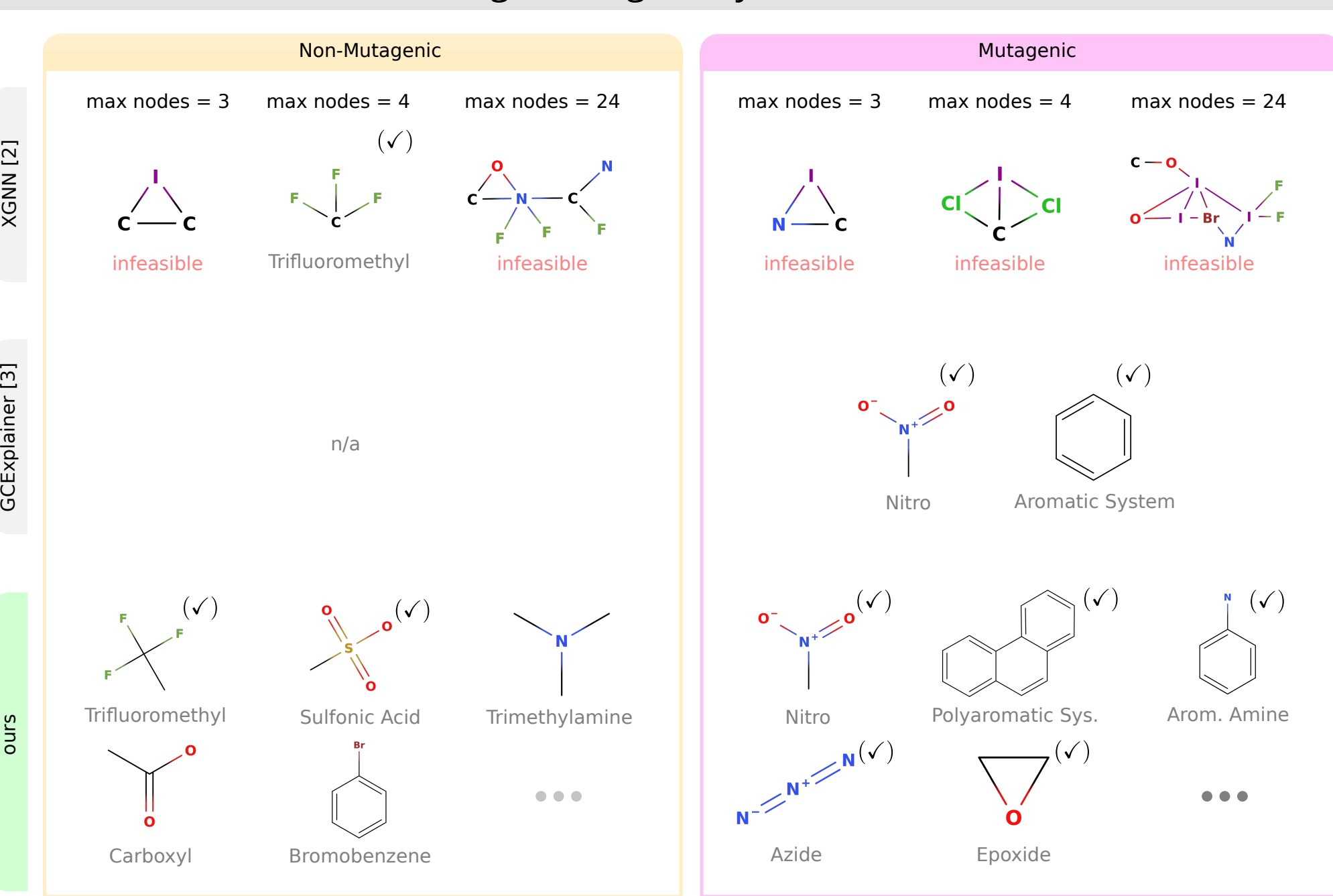

Mutag - Mutagenicity Classification

(✓) consistent with hypotheses previously published by Kazius *et al.* [4].

## ...Rediscover Known Structure Property Relationships From Chemistry Literature

[1] Teufel, Torresi, Reiser, Friederich. *MEGAN: Multi-Explanation Graph Attention Network*. xAI Conference 2023. CCIS Volume 1902 pp 228-360
[2] Yuan, Tang, Hu, Ji. *XGNN: Towards Model-Level Explanations of Graph Neural Networks*. KDD Conference 2020. SIGKDD Volume 26 p 430-438
[3] Magister, Kazhdan, Singh, Lio. *GCExplainer: Human-in-the-Loop Concept-based Explanations for Graph Neural Networks*. Arxiv 2021
[4] Kazius, McGuire, Bursi. *Derivation and Validation of Toxicophores for Mutagenicity Prediction*. J. Med. Chem 2005. Volume 48 p 312-320

