# OpenReview forum: "Finding Structure-Property Relationships for Molecular Property Predictions with Globally Explainable AI"
_ICML.cc/2024/Workshop/ML4LMS — ML4LMS Poster_

### Official Review · Reviewer_gj2D · 2024-06-11
**Review of Submission22**

**Rating:** 7
**Confidence:** 4

**Review:**

The poster presents a workflow for identifying structure-property relationships in molecular property predictions using XAI. Specifically, the proposed workflow integrates the MEGAN approach to generate local explanation masks, employs latent space clustering to identify structural similarities, and uses cluster analysis to associate structural motifs with properties. Additionally, the workflow leverages GPT-4 to generate causal hypotheses based on prototype graphs. Overall, the poster is well-organized and provides application examples (AqSolDB and Mutag). The topic is meaningful and the work may stimulate further discussion and exploration. Here are a few comments and questions to consider and address:

1. While HDBSCAN is used for clustering, it is unclear how the quality or validity of these clusters is assessed. The authors may conside including some metrics to further strengthen the cluster analysis.

2. What is the prediction accuracy of the examples illustrated in the poster (compared to other benchmarking models)? If the prediction accuracy is not satisfactory, how does this affect the explanatory power of the motifs contributing to specific prediction tasks? Additionally, how stable are these explanations? Could the authors also comment on the motifs found in Mutag that are not consistent with previously published papers?

3. The audience would benefit from more details on the implementation and training processes. For instance, while the genetic algorithm for finding prototype graphs is mentioned, more information on its selection criteria would enhance clarity. Additionally, the narrative context for generating causal hypotheses using GPT-4 is unclear and would benefit from further explanation.

---

### Official Review · Reviewer_61ob · 2024-06-11
**The work presents a methodological framework that integrates the MEGAN model, latent space clustering, and a prediction network to uncover causal relationships and provide interpretable predictions.**

**Rating:** 7
**Confidence:** 4

**Review:**

The study leverages advanced machine learning techniques, specifically tailored towards improving the interpretability of neural network predictions in chemical informatics. The integration of causal hypotheses within the model structure is notable and promises to enhance the understanding of complex biological and chemical datasets. However, the lack of extensive validation metrics and demonstration of the model's performance across diverse chemical datasets limits its generalisability.